# Quaternion Knowledge Graph Embeddings

**Shuai Zhang**[†*]**, Yi Tay**[ψ*]**, Lina Yao**[†]**, Qi Liu**[φ]
[†] University of New South Wales
[ψ]Nanyang Technological University, [φ]University of Oxford

## Abstract

In this work, we move beyond the traditional complex-valued representations, introducing more expressive hypercomplex representations to model entities and relations for knowledge graph embeddings. More specifically, quaternion embeddings, hypercomplex-valued embeddings with three imaginary components, are utilized to represent entities. Relations are modelled as rotations in the quaternion space. The advantages of the proposed approach are: (1) Latent inter-dependencies (between all components) are aptly captured with Hamilton product, encouraging a more compact interaction between entities and relations; (2) Quaternions enable expressive rotation in four-dimensional space and have more degree of freedom than rotation in complex plane; (3) The proposed framework is a generalization of ComplEx on hypercomplex space while offering better geometrical interpretations, concurrently satisfying the key desiderata of relational representation learning (i.e., modeling symmetry, anti-symmetry and inversion). Experimental results demonstrate that our method achieves state-of-the-art performance on four well-established knowledge graph completion benchmarks.

## 1 Introduction

Knowledge graphs (KGs) live at the heart of many semantic applications (e.g., question answering, search, and natural language processing). KGs enable not only powerful relational reasoning but also the ability to learn structural representations. Reasoning with KGs have been an extremely productive research direction, with many innovations leading to improvements to many downstream applications. However, real-world KGs are usually incomplete. As such, completing KGs and predicting missing links between entities have gained growing interest. Learning low-dimensional representations of entities and relations for KGs is an effective solution for this task.

Learning KG embeddings in the complex space $\mathbb{C}$ has been proven to be a highly effective inductive bias, largely owing to its intrinsic asymmetrical properties. This is demonstrated by the ComplEx embedding method which infers new relational triplets with the asymmetrical Hermitian product.

In this paper, we move beyond complex representations, exploring hypercomplex space for learning KG embeddings. More concretely, quaternion embeddings are utilized to represent entities and relations. Each quaternion embedding is a vector in the hypercomplex space $\mathbb{H}$ with three imaginary components $\mathbf{i}, \mathbf{j}, \mathbf{k}$, as opposed to the standard complex space $\mathbb{C}$ with a single real component $r$ and imaginary component $\mathbf{i}$. We propose a new scoring function, where the head entity $Q_h$ is rotated by the relational quaternion embedding through Hamilton product. This is followed by a quaternion inner product with the tail entity $Q_t$.

There are numerous benefits of this formulation. (1) The Hamilton operator provides a greater extent of expressiveness compared to the complex Hermitian operator and the inner product in Euclidean space. The Hamilton operator forges inter-latent interactions between all of $r, \mathbf{i}, \mathbf{j}, \mathbf{k}$, resulting in

---

[*]Equal contribution.

a highly expressive model. (2) Quaternion representations are highly desirable for parameterizing smooth rotation and spatial transformations in vector space. They are generally considered robust to sheer/scaling noise and perturbations (i.e., numerically stable rotations) and avoid the problem of Gimbal locks. Moreover, quaternion rotations have two planes of rotation[2] while complex rotations only work on single plane, giving the model more degrees of freedom. (3) Our QuatE framework subsumes the ComplEx method, concurrently inheriting its attractive properties such as its ability to model symmetry, anti-symmetry, and inversion. (4) Our model can maintain equal or even less parameterization, while outperforming previous work.

Experimental results demonstrate that our method achieves state-of-the-art performance on four well-established knowledge graph completion benchmarks (WN18, FB15K, WN18RR, and FB15K-237).

## 2 Related Work

Knowledge graph embeddings have attracted intense research focus in recent years, and a myriad of embedding methodologies have been proposed. We roughly divide previous work into translational models and semantic matching models based on the scoring function, i.e. the composition over head & tail entities and relations.

Translational methods popularized by TransE [Bordes et al., 2013] are widely used embedding methods, which interpret relation vectors as translations in vector space, i.e., $head + relation \approx tail$. A number of models aiming to improve TransE are proposed subsequently. TransH [Wang et al., 2014] introduces relation-specific hyperplanes with a normal vector. TransR [Lin et al., 2015] further introduces relation-specific space by modelling entities and relations in distinct space with a shared projection matrix. TransD [Ji et al., 2015] uses independent projection vectors for each entity and relation and can reduce the amount of calculation compared to TransR. TorusE [Ebisu and Ichise, 2018] defines embeddings and distance function in a compact Lie group, torus, and shows better accuracy and scalability. The recent state-of-the-art, RotatE [Sun et al., 2019], proposes a rotation-based translational method with complex-valued embeddings.

On the other hand, semantic matching models include bilinear models such as RESCAL [Nickel et al., 2011], DistMult [Yang et al., 2014], HolE [Nickel et al., 2016], and ComplEx [Trouillon et al., 2016], and neural-network-based models. These methods measure plausibility by matching latent semantics of entities and relations. In RESCAL, each relation is represented with a square matrix, while DistMult replaces it with a diagonal matrix in order to reduce the complexity. SimplE [Kazemi and Poole, 2018] is also a simple yet effective bilinear approach for knowledge graph embedding. HolE explores the holographic reduced representations and makes use of circular correlation to capture rich interactions between entities. ComplEx embeds entities and relations in complex space and utilizes Hermitian product to model the antisymmetric patterns, which has shown to be immensely helpful in learning KG representations. The scoring function of ComplEx is isomorphic to that of HolE [Trouillon and Nickel, 2017]. Neural networks based methods have also been adopted, e.g., Neural Tensor Network [Socher et al., 2013] and ER-MLP [Dong et al., 2014] are two representative neural network based methodologies. More recently, convolution neural networks [Dettmers et al., 2018], graph convolutional networks [Schlichtkrull et al., 2018], and deep memory networks [Wang et al., 2018] also show promising performance on this task.

Different from previous work, QuatE takes the advantages (e.g., its geometrical meaning and rich representation capability, etc.) of quaternion representations to enable rich and expressive semantic matching between head and tail entities, assisted by relational rotation quaternions. Our framework subsumes DistMult and ComplEx, with the capability to generalize to more advanced hypercomplex spaces. QuatE utilizes the concept of geometric rotation. Unlike the RotatE which has only one plane of rotation, there are two planes of rotation in QuatE. QuatE is a semantic matching model while RotatE is a translational model. We also point out that the composition property introduced in TransE and RotatE can have detrimental effects on the KG embedding task.

Quaternion is a hypercomplex number system firstly described by Hamilton [Hamilton, 1844] with applications in a wide variety of areas including astronautics, robotics, computer visualisation, animation and special effects in movies, and navigation. Lately, Quaternions have attracted attention in the field of machine learning. Quaternion recurrent neural networks (QRNNs) obtain better performance with

fewer number of free parameters than traditional RNNs on the phoneme recognition task. Quaternion representations are also useful for enhancing the performance of convolutional neural networks on multiple tasks such as automatic speech recognition [Parcollet et al.] and image classification [Gaudet and Maida, 2018, Parcollet et al., 2018a]. Quaternion multiplayer perceptron [Parcollet et al., 2016] and quaternion autoencoders [Parcollet et al., 2017] also outperform standard MLP and autoencoder. In a nutshell, the major motivation behind these models is that quaternions enable the neural networks to code latent inter- and intra-dependencies between multidimensional input features, thus, leading to more compact interactions and better representation capability.

## 3 Hamilton's Quaternions

Quaternion [Hamilton, 1844] is a representative of hypercomplex number system, extending traditional complex number system to four-dimensional space. A quaternion $Q$ consists of one real component and three imaginary components, defined as $Q = a + b\mathbf{i} + c\mathbf{j} + d\mathbf{k}$, where $a, b, c, d$ are real numbers and $\mathbf{i}, \mathbf{j}, \mathbf{k}$ are imaginary units. $\mathbf{i}, \mathbf{j}$ and $\mathbf{k}$ are square roots of $-1$, satisfying the Hamilton's rules: $\mathbf{i}^2 = \mathbf{j}^2 = \mathbf{k}^2 = \mathbf{ijk} = -1$. More useful relations can be derived based on these rules, such as $\mathbf{ij = k}$, $\mathbf{ji = -k}$, $\mathbf{jk=i}$, $\mathbf{ki=j}$, $\mathbf{kj=-i}$ and $\mathbf{ik=-j}$. Figure 1(b) shows the quaternion imaginary units product. Apparently, the multiplication between imaginary units is non-commutative. Some widely used operations of quaternion algebra $\mathbb{H}$ are introduced as follows:

**Conjugate**: The conjugate of a quaternion $Q$ is defined as $\bar{Q} = a - b\mathbf{i} - c\mathbf{j} - d\mathbf{k}$.

**Norm**: The norm of a quaternion is defined as $|Q| = \sqrt{a^2 + b^2 + c^2 + d^2}$.

**Inner Product**: The quaternion inner product between $Q_1 = a_1 + b_1\mathbf{i} + c_1\mathbf{j} + d_1\mathbf{k}$ and $Q_2 = a_2 + b_2\mathbf{i} + c_2\mathbf{j} + d_2\mathbf{k}$ is obtained by taking the inner products between corresponding scalar and imaginary components and summing up the four inner products:

$$Q_1 \cdot Q_2 = \langle a_1, a_2 \rangle + \langle b_1, b_2 \rangle + \langle c_1, c_2 \rangle + \langle d_1, d_2 \rangle \tag{1}$$

**Hamilton Product (Quaternion Multiplication)**: The Hamilton product is composed of all the standard multiplications of factors in quaternions and follows the distributive law, defined as:

$$\begin{aligned} Q_1 \otimes Q_2 = &(a_1a_2 - b_1b_2 - c_1c_2 - d_1d_2) + (a_1b_2 + b_1a_2 + c_1d_2 - d_1c_2)\mathbf{i} \\ &+ (a_1c_2 - b_1d_2 + c_1a_2 + d_1b_2)\mathbf{j} + (a_1d_2 + b_1c_2 - c_1b_2 + d_1a_2)\mathbf{k}\text{'} \end{aligned} \tag{2}$$

which determines another quaternion. Hamilton product is not commutative. Spatial rotations can be modelled with quaternions Hamilton product. Multiplying a quaternion, $Q_2$, by another quaternion $Q_1$, has the effect of scaling $Q_1$ by the magnitude of $Q_2$ followed by a special type of rotation in four dimensions. As such, we can also rewrite the above equation as:

$$Q_1 \otimes Q_2 = Q_1 \otimes |Q_2|\left(\frac{Q_2}{|Q_2|}\right) \tag{3}$$

## 4 Method

### 4.1 Quaternion Representations for Knowledge Graph Embeddings

Suppose that we have a knowledge graph $\mathcal{G}$ consisting of $N$ entities and $M$ relations. $\mathcal{E}$ and $\mathcal{R}$ denote the sets of entities and relations, respectively. The training set consists of triplets $(h, r, t)$, where $h, t \in \mathcal{E}$ and $r \in \mathcal{R}$. We use $\Omega$ and $\Omega' = \mathcal{E} \times \mathcal{R} \times \mathcal{E} - \Omega$ to denote the set of observed triplets and the set of unobserved triplets, respectively. $Y_{hrt} \in \{-1, 1\}$ represents the corresponding label of the triplet $(h, r, t)$. The goal of knowledge graph embeddings is to embed entities and relations to a continuous low-dimensional space, while preserving graph relations and semantics.

In this paper, we propose learning effective representations for entities and relations with quaternions. We leverage the expressive rotational capability of quaternions. Unlike RotatE which has only one plane of rotation (i.e., complex plane, shown in Figure 1(a)), QuatE has two planes of rotation. Compared to Euler angles, quaternion can avoid the problem of gimbal lock (loss of one degree of freedom). Quaternions are also more efficient and numerically stable than rotation matrices. The proposed method can be summarized into two steps: (1) rotate the head quaternion using the unit relation quaternion; (2) take the quaternion inner product between the rotated head quaternion and

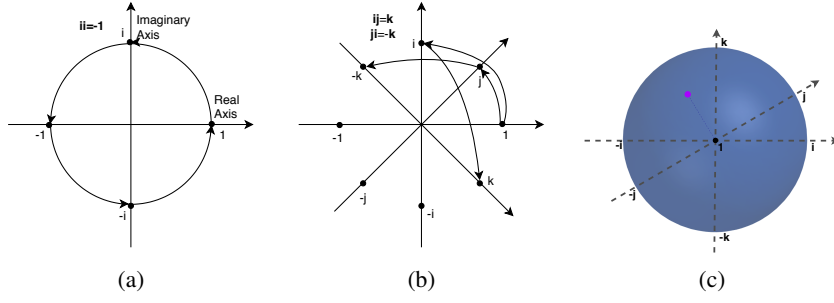

Figure 1: (a) Complex plane; (b) Quaternion units product; (c) sterographically projected hypersphere in 3D space. The purple dot indicates the position of the unit quaternion.

the tail quaternion to score each triplet. If a triplet exists in the KG, the model will rotate the head entity with the relation to make the angle between head and tail entities smaller so the product can be maximized. Otherwise, we can make the head and tail entity be orthogonal so that their product becomes zero.

**Quaternion Embeddings of Knowledge Graphs**  More specifically, we use a quaternion matrix $Q \in \mathbb{H}^{N \times k}$ to denote the entity embeddings and $W \in \mathbb{H}^{M \times k}$ to denote the relation embeddings, where $k$ is the dimension of embeddings. Given a triplet $(h, r, t)$, the head entity $h$ and the tail entity $t$ correspond to $Q_h = \{a_h + b_h\mathbf{i} + c_h\mathbf{j} + d_h\mathbf{k} : a_h, b_h, c_h, d_h \in \mathbb{R}^k\}$ and $Q_t = \{a_t + b_t\mathbf{i} + c_t\mathbf{j} + d_t\mathbf{k} : a_t, b_t, c_t, d_t \in \mathbb{R}^k\}$, respectively, while the relation $r$ is represented by $W_r = \{a_r + b_r\mathbf{i} + c_r\mathbf{j} + d_r\mathbf{k} : a_r, b_r, c_r, d_r \in \mathbb{R}^k\}$.

**Hamilton-Product-Based Relational Rotation**  We first normalize the relation quaternion $W_r$ to a unit quaternion $W_r^{\triangleleft} = p + q\mathbf{i} + u\mathbf{j} + v\mathbf{k}$ to eliminate the scaling effect by dividing $W_r$ by its norm:

$$W_r^{\triangleleft}(p, q, u, v) = \frac{W_r}{|W_r|} = \frac{a_r + b_r\mathbf{i} + c_r\mathbf{j} + d_r\mathbf{k}}{\sqrt{a_r^2 + b_r^2 + c_r^2 + d_r^2}} \tag{4}$$

We visualize a unit quaternion in Figure 1(c) by projecting it into a 3D space. We keep the unit hypersphere which passes through $\mathbf{i}, \mathbf{j}, \mathbf{k}$ in place. The unit quaternion can be project in, on, or out of the unit hypersphere depending on the value of the real part.

Secondly, we rotate the head entity $Q_h$ by doing Hamilton product between it and $W_r^{\triangleleft}$:

$$\begin{aligned}
Q_h'(a_h', b_h', c_h', d_h') = Q_h \otimes W_r^{\triangleleft} = {}&(a_h \circ p - b_h \circ q - c_h \circ u - d_h \circ v) \\
&+ (a_h \circ q + b_h \circ p + c_h \circ v - d_h \circ u)\mathbf{i} \\
&+ (a_h \circ u - b_h \circ v + c_h \circ p + d_h \circ q)\mathbf{j} \\
&+ (a_h \circ v + b_h \circ u - c_h \circ q + d_h \circ p)\mathbf{k}
\end{aligned} \tag{5}$$

where $\circ$ denotes the element-wise multiplication between two vectors. Right-multiplication by a unit quaternion is a right-isoclinic rotation on Quaternion $Q_h$. We can also swap $Q_h$ and $W_r^{\triangleleft}$ and do a left-isoclinic rotation, which does not fundamentally change the geometrical meaning. Isoclinic rotation is a special case of double plane rotation where the angles for each plane are equal.

**Scoring Function and Loss**  We apply the quaternion inner product as the scoring function:

$$\phi(h, r, t) = Q_h' \cdot Q_t = \langle a_h', a_t \rangle + \langle b_h', b_t \rangle + \langle c_h', c_t \rangle + \langle d_h', d_t \rangle \tag{6}$$

Following Trouillon et al. [2016], we formulate the task as a classification problem, and the model parameters are learned by minimizing the following regularized logistic loss:

$$L(Q, W) = \sum_{r(h,t) \in \Omega \cup \Omega^-} \log(1 + \exp(-Y_{hrt}\phi(h, r, t))) + \lambda_1 \parallel Q \parallel_2^2 + \lambda_2 \parallel W \parallel_2^2 \tag{7}$$

Here we use the $\ell_2$ norm with regularization rates $\lambda_1$ and $\lambda_2$ to regularize $Q$ and $W$, respectively. $\Omega^-$ is sampled from the unobserved set $\Omega'$ using negative sampling strategies such as uniform sampling, bernoulli sampling [Wang et al., 2014], and adversarial sampling [Sun et al., 2019]. Note that the loss function is in Euclidean space, as we take the summation of all components when computing the scoring function in Equation (6). We utilise Adagrad [Duchi et al., 2011] for optimization.

Table 1: Scoring functions of state-of-the-art knowledge graph embedding models, along with their parameters, time complexity. "$\star$" denotes the circular correlation operation; "$\circ$" denotes Hadmard (or element-wise) product. "$\otimes$" denotes Hamilton product.

| Model | Scoring Function | Parameters | $\mathcal{O}_{time}$ |
|---|---|---|---|
| TransE | $\| (Q_h + W_r) - Q_t \|$ | $Q_h, W_r, Q_t \in \mathbb{R}^k$ | $\mathcal{O}(k)$ |
| HolE | $\langle W_r, Q_h \star Q_t \rangle$ | $Q_h, W_r, Q_t \in \mathbb{R}^k$ | $\mathcal{O}(k \log(k))$ |
| DistMult | $\langle W_r, Q_h, Q_t \rangle$ | $Q_h, W_r, Q_t \in \mathbb{R}^k$ | $\mathcal{O}(k)$ |
| ComplEx | $\mathrm{Re}(\langle W_r, Q_h, \bar{Q}_t \rangle)$ | $Q_h, W_r, Q_t \in \mathbb{C}^k$ | $\mathcal{O}(k)$ |
| RotatE | $\| Q_h \circ W_r - Q_t \|$ | $Q_h, W_r, Q_t \in \mathbb{C}^k, |W_{ri}| = 1$ | $\mathcal{O}(k)$ |
| TorusE | $min_{(x,y) \in ([Q_h]+[Q_h]) \times [W_r]} \| x - y \|$ | $[Q_h], [W_r], [Q_t] \in \mathbb{T}^k$ | $\mathcal{O}(k)$ |
| **QuatE** | $Q_h \otimes W_r^{\triangleleft} \cdot Q_t$ | $Q_h, W_r, Q_t \in \mathbb{H}^k$ | $\mathcal{O}(k)$ |

**Initialization**  For parameters initilaization, we can adopt the initialization algorithm in [Parcollet et al., 2018b] tailored for quaternion-valued networks to speed up model efficiency and convergence [Glorot and Bengio, 2010]. The initialization of entities and relations follows the rule:

$$w_{real} = \varphi \cos(\theta), w_i = \varphi Q_{img_{\mathbf{i}}}^{\triangleleft} \sin(\theta), w_j = \varphi Q_{img_{\mathbf{j}}}^{\triangleleft} \sin(\theta), w_k = \varphi Q_{img_{\mathbf{k}}}^{\triangleleft} \sin(\theta), \qquad (8)$$

where $w_{real}, w_i, w_j, w_k$ denote the scalar and imaginary coefficients, respectively. $\theta$ is randomly generated from the interval $[-\pi, \pi]$. $Q_{img}^{\triangleleft}$ is a normalized quaternion, whose scalar part is zero. $\varphi$ is randomly generated from the interval $[-\frac{1}{\sqrt{2k}}, \frac{1}{\sqrt{2k}}]$, reminiscent to the He initialization [He et al., 2015]. This initialization method is optional.

## 4.2  Discussion

Table 1 summarizes several popular knowledge graph embedding models, including scoring functions, parameters, and time complexities. TransE, HolE, and DistMult use Euclidean embeddings, while ComplEx and RotatE operate in the complex space. In contrast, our model operates in the quaternion space.

**Capability in Modeling Symmetry, Antisymmetry and Inversion**. The flexibility and representational power of quaternions enable us to model major relation patterns at ease. Similar to ComplEx, our model can model both symmetry $(r(x,y) \Rightarrow r(y,x))$ and antisymmetry $(r(x,y) \Rightarrow \neg r(y,x))$ relations. The symmetry property of QuatE can be proved by setting the imaginary parts of $W_r$ to zero. One can easily check that the scoring function is antisymmetric when the imaginary parts are nonzero.

As for the inversion pattern $(r_1(x,y) \Rightarrow r_2(y,x))$ , we can utilize the conjugation of quaternions. Conjugation is an involution and is its own inverse. One can easily check that:

$$Q_h \otimes W_r^{\triangleleft} \cdot Q_t = Q_t \otimes \bar{W}_r^{\triangleleft} \cdot Q_h \qquad (9)$$

The detailed proof of antisymmetry and inversion can be found in the appendix.

Composition patterns are commonplace in knowledge graphs [Lao et al., 2011, Neelakantan et al., 2015]. Both transE and RotatE have fixed composition methods [Sun et al., 2019]. TransE composes two relations using the addition $(r_1 + r_2)$ and RotatE uses the Hadamard product $(r_1 \circ r_2)$. We argue that it is unreasonable to fix the composition patterns, as there might exist multiple composition patterns even in a single knowledge graph. For example, suppose there are three persons "$x, y, z$". If $y$ is the elder sister (denoted as $r_1$) of $x$ and $z$ is the elder brother (denoted as $r_2$) of $y$, we can easily infer that $z$ is the elder brother of $x$. The relation between $z$ and $x$ is $r_2$ instead of $r_1 + r_2$ or $r_1 \circ r_2$, violating the two composition methods of TransE and RotatE. In QuatE, the composition patterns are not fixed. The relation between $z$ and $x$ is not only determined by relations $r_1$ and $r_2$ but also simultaneously influenced by entity embeddings.

**Connection to DistMult and ComplEx.** Quaternions have more degrees of freedom compared to complex numbers. Here we show that the QuatE framework can be seen as a generalization of ComplEx. If we set the coefficients of the imaginary units **j** and **k** to zero, we get complex embeddings as in ComplEx and the Hamilton product will also degrade to complex number multiplication. We

Table 2: Statistics of the data sets used in this paper.

| Dataset | N | M | #training | #validation | #test | avg. #degree |
|---------|-----|------|-----------|-------------|-------|--------------|
| WN18 | 40943 | 18 | 141442 | 5000 | 5000 | 3.45 |
| WN18RR | 40943 | 11 | 86835 | 3034 | 3134 | 2.19 |
| FB15K | 14951 | 1345 | 483142 | 50000 | 59071 | 32.31 |
| FB15K-237 | 14541 | 237 | 272115 | 17535 | 20466 | 18.71 |

further remove the normalization of the relational quaternion, obtaining the following equation:

$$\phi(h, r, t) = Q_h \otimes W_r \cdot Q_t = (a_h + b_h \mathbf{i}) \otimes (a_r + b_r \mathbf{i}) \cdot (a_t + b_t \mathbf{i})$$
$$= [(a_h \circ a_r - b_h \circ b_r) + (a_h \circ b_r + b_h \circ a_r)\mathbf{i}] \cdot (a_t + b_t \mathbf{i}) \qquad (10)$$
$$= \langle a_r, a_h, a_t \rangle + \langle a_r, b_h, b_t \rangle + \langle b_r, a_h, b_t \rangle - \langle b_r, b_h, a_t \rangle$$

where $\langle a, b, c \rangle = \sum_k a_k b_k c_k$ denotes standard component-wise multi-linear dot product. Equation 10 recovers the form of ComplEx. This framework brings another mathematical interpretation for ComplEx instead of just taking the real part of the Hermitian product. Another interesting finding is that Hermitian product is not necessary to formulate the scoring function of ComplEx.

If we remove the imaginary parts of all quaternions and remove the normalization step, the scoring function becomes $\phi(h, r, t) = \langle a_h, a_r, a_t \rangle$, degrading to DistMult in this case.

## 5 Experiments and Results

### 5.1 Experimental Setup

**Datasets Description:** We conducted experiments on four widely used benchmarks, WN18, FB15K, WN18RR and FB15K-237, of which the statistics are summarized in Table 2. WN18 [Bordes et al., 2013] is extracted from WordNet[3], a lexical database for English language, where words are interlinked by means of conceptual-semantic and lexical relations. WN18RR [Dettmers et al., 2018] is a subset of WN18, with inverse relations removed. FB15K [Bordes et al., 2013] contains relation triples from Freebase, a large tuple database with structured general human knowledge. FB15K-237 [Toutanova and Chen, 2015] is a subset of FB15K, with inverse relations removed.

**Evaluation Protocol:** Three popular evaluation metrics are used, including Mean Rank (MR), Mean Reciprocal Rank (MRR), and Hit ratio with cut-off values $n = 1, 3, 10$. MR measures the average rank of all correct entities with a lower value representing better performance. MRR is the average inverse rank for correct entities. Hit@n measures the proportion of correct entities in the top $n$ entities. Following Bordes et al. [2013], filtered results are reported to avoid possibly flawed evaluation.

**Baselines:** We compared QuatE with a number of strong baselines. For *Translational Distance Models*, we reported TransE [Bordes et al., 2013] and two recent extensions, TorusE [Ebisu and Ichise, 2018] and RotatE [Sun et al., 2019]; For *Semantic Matching Models*, we reported DistMult [Yang et al., 2014], HolE [Nickel et al., 2016], ComplEx [Trouillon et al., 2016] , SimplE [Kazemi and Poole, 2018], ConvE [Dettmers et al., 2018], R-GCN [Schlichtkrull et al., 2018], and KNGE (ConvE based) [Wang et al., 2018].

**Implementation Details:** We implemented our model using pytorch[4] and tested it on a single GPU. The hyper-parameters are determined by grid search. The best models are selected by early stopping on the validation set. In general, the embedding size $k$ is tuned amongst $\{50, 100, 200, 250, 300\}$. Regularization rate $\lambda_1$ and $\lambda_2$ are searched in $\{0, 0.01, 0.05, 0.1, 0.2\}$. Learning rate is fixed to 0.1 without further tuning. The number of negatives ($\#neg$) per training sample is selected from $\{1, 5, 10, 20\}$. We create 10 batches for all the datasets. For most baselines, we report the results in the original papers, and exceptions are provided with references. For RotatE (without self-adversarial negative sampling), we use the best hyper-parameter settings provided in the paper to reproduce the results. We also report the results of RotatE with self-adversarial negative sampling and denote it as a-RotatE. Note that we report three versions of QuatE: including QuatE with/without type constraints, QuatE with N3 regularization and reciprocal learning. Self-adversarial negative sampling [Sun et al., 2019] is not used for QuatE. All hyper-parameters of QuatE are provided in the appendix.

Table 3: Link prediction results on WN18 and FB15K. Best results are in bold and second best results are underlined. [†]: Results are taken from [Nickel et al., 2016]; [◇]: Results are taken from [Kadlec et al., 2017]; [∗]: Results are taken from [Sun et al., 2019]. a-RotatE denotes RotatE with self-adversarial negative sampling. [QuatE$^1$]: without type constraints; [QuatE$^2$]: with N3 regularization and reciprocal learning; [QuatE$^3$]: with type constraints.

| Model | WN18 | | | | | FB15K | | | | |
|---|---|---|---|---|---|---|---|---|---|---|
| | MR | MRR | Hit@10 | Hit@3 | Hit@1 | MR | MRR | Hit@10 | Hit@3 | Hit@1 |
| TransE† | - | 0.495 | 0.943 | 0.888 | 0.113 | - | 0.463 | 0.749 | 0.578 | 0.297 |
| DistMult◇ | 655 | 0.797 | 0.946 | - | - | 42.2 | 0.798 | 0.893 | - | - |
| HolE | - | 0.938 | 0.949 | 0.945 | 0.930 | - | 0.524 | 0.739 | 0.759 | 0.599 |
| ComplEx | - | 0.941 | 0.947 | 0.945 | 0.936 | - | 0.692 | 0.840 | 0.759 | 0.599 |
| ConvE | 374 | 0.943 | 0.956 | 0.946 | 0.935 | 51 | 0.657 | 0.831 | 0.723 | 0.558 |
| R-GCN+ | - | 0.819 | **0.964** | 0.929 | 0.697 | - | 0.696 | 0.842 | 0.760 | 0.601 |
| SimplE | - | 0.942 | 0.947 | 0.944 | 0.939 | - | 0.727 | 0.838 | 0.773 | 0.660 |
| NKGE | 336 | 0.947 | 0.957 | 0.949 | 0.942 | 56 | 0.73 | 0.871 | 0.790 | 0.650 |
| TorusE | - | 0.947 | 0.954 | 0.950 | 0.943 | - | 0.733 | 0.832 | 0.771 | 0.674 |
| RotatE | 184 | 0.947 | 0.961 | 0.953 | 0.938 | 32 | 0.699 | 0.872 | 0.788 | 0.585 |
| a-RotatE∗ | 309 | 0.949 | 0.959 | 0.952 | 0.944 | 40 | 0.797 | 0.884 | 0.830 | 0.746 |
| **QuatE**$^1$ | 388 | 0.949 | 0.960 | **0.954** | 0.941 | 41 | 0.770 | 0.878 | 0.821 | 0.700 |
| **QuatE**$^2$ | - | **0.950** | 0.962 | **0.954** | 0.944 | - | **0.833** | **0.900** | **0.859** | **0.800** |
| **QuatE**$^3$ | **162** | **0.950** | 0.959 | **0.954** | **0.945** | **17** | 0.782 | **0.900** | 0.835 | 0.711 |

Table 4: Link prediction results on WN18RR and FB15K-237. [†]: Results are taken from [Nguyen et al., 2017]; [◇]: Results are taken from [Dettmers et al., 2018]; [∗]: Results are taken from [Sun et al., 2019].

| Model | WN18RR | | | | | FB15K-237 | | | | |
|---|---|---|---|---|---|---|---|---|---|---|
| | MR | MRR | Hit@10 | Hit@3 | Hit@1 | MR | MRR | Hit@10 | Hit@3 | Hit@1 |
| TransE † | 3384 | 0.226 | 0.501 | - | - | 357 | 0.294 | 0.465 | - | - |
| DistMult◇ | 5110 | 0.43 | 0.49 | 0.44 | 0.39 | 254 | 0.241 | 0.419 | 0.263 | 0.155 |
| ComplEx◇ | 5261 | 0.44 | 0.51 | 0.46 | 0.41 | 339 | 0.247 | 0.428 | 0.275 | 0.158 |
| ConvE◇ | 4187 | 0.43 | 0.52 | 0.44 | 0.40 | 244 | 0.325 | 0.501 | 0.356 | 0.237 |
| R-GCN+ | - | - | - | - | - | - | 0.249 | 0.417 | 0.264 | 0.151 |
| NKGE | 4170 | 0.45 | 0.526 | 0.465 | 0.421 | 237 | 0.33 | 0.510 | 0.365 | 0.241 |
| RotatE∗ | 3277 | 0.470 | 0.565 | 0.488 | 0.422 | 185 | 0.297 | 0.480 | 0.328 | 0.205 |
| a-RotatE∗ | 3340 | 0.476 | 0.571 | 0.492 | 0.428 | 177 | 0.338 | 0.533 | 0.375 | 0.241 |
| **QuatE**$^1$ | 3472 | 0.481 | 0.564 | 0.500 | 0.436 | 176 | 0.311 | 0.495 | 0.342 | 0.221 |
| **QuatE**$^2$ | - | 0.482 | 0.572 | 0.499 | 0.436 | - | **0.366** | **0.556** | **0.401** | **0.271** |
| **QuatE**$^3$ | **2314** | **0.488** | **0.582** | **0.508** | **0.438** | **87** | 0.348 | 0.550 | 0.382 | 0.248 |

## 5.2 Results

The empirical results on four datasets are reported in Table 3 and Table 4. QuatE performs extremely competitively compared to the existing state-of-the-art models across all metrics. As a quaternion-valued method, QuatE outperforms the two representative complex-valued models ComplEx and RotatE. The performance gains over RotatE also confirm the advantages of quaternion rotation over rotation in the complex plane.

Table 5: MRR for the models tested on each relation of WN18RR.

| Relation Name | RotatE | **QuatE**$^3$ |
|---|---|---|
| hypernym | 0.148 | **0.173** |
| derivationally_related_form | 0.947 | **0.953** |
| instance_hypernym | 0.318 | **0.364** |
| also_see | 0.585 | **0.629** |
| member_meronym | **0.232** | **0.232** |
| synset_domain_topic_of | 0.341 | **0.468** |
| has_part | 0.184 | **0.233** |
| member_of_domain_usage | 0.318 | **0.441** |
| member_of_domain_region | **0.200** | 0.193 |
| verb_group | **0.943** | 0.924 |
| similar_to | **1.000** | **1.000** |

On the WN18 dataset, QuatE outperforms all the baselines on all metrics except Hit@10. R-GCN+ achieves high value on Hit@10, yet is surpassed by most models on the other four metrics. The four recent models NKGE, TorusE, RotaE, and a-RotatE achieves comparable results. QuatE also achieves the best results on the FB15K dataset, while the second best results scatter amongst RotatE, a-RotatE and DistMult. We are well-aware of the good results of DistMult reported in [Kadlec et al., 2017], yet they used a very large negative sampling size (i.e., 1000, 2000). The results also demonstrate that

Table 7: Analysis on different variants of scoring function. Same hyperparameters settings as QuatE[3] are used.

| Analysis | WN18 | | FB15K | | WN18RR | | FB15K-237 | |
|---|---|---|---|---|---|---|---|---|
| | MRR | Hit@10 | MRR | Hit@10 | MRR | Hit@10 | MRR | Hit@10 |
| $Q_h \otimes W_r \cdot Q_t$ | 0.936 | 0.951 | 0.686 | 0.866 | 0.415 | 0.482 | 0.272 | 0.463 |
| $W_r \cdot (Q_h \otimes Q_t)$ | 0.784 | 0.945 | 0.599 | 0.809 | 0.401 | 0.471 | 0.263 | 0.446 |
| $(Q_h \otimes W_r^{\triangleleft}) \cdot (Q_t \otimes V_r^{\triangleleft})$ | 0.947 | 0.958 | 0.787 | 0.889 | 0.477 | 0.563 | 0.344 | 0.539 |

QuatE can effectively capture the symmetry, antisymmetry and inversion patterns since they account for a large portion of the relations in these two datasets.

As shown in Table 4, QuatE achieves a large performance gain over existing state-of-the-art models on the two datasets where trivial inverse relations are removed. On WN18RR in which there are a number of symmetry relations, a-RotatE is the second best, while other baselines are relatively weaker. The key competitors on the dataset FB15K-237 where a large number of composition patterns exist are NKGE and a-RotatE. Table 5 summarizes the MRR for each relation on WN18RR, confirming the superior representation capability of quaternion in modelling different types of relation. Methods with fixed composition patterns such as TransE and RotatE are relatively weak at times.

We can also apply N3 regularization and reciprocal learning approaches [Lacroix et al., 2018] to QuatE. Results are shown in Table 3 and Table 4 as QuatE[2]. It is observed that using N3 and reciprocal learning could boost the performances greatly, especially on FB15K and FB15K-237. We found that the N3 regularization method can reduce the norm of relations and entities embeddings so that we do not apply relation normalization here. However, same as the method in [Lacroix et al., 2018], QuatE[2] requires a large embedding dimension.

## 5.3 Model Analysis

**Number of Free Parameters Comparison**. Table 6 shows the amount of parameters comparison between QuatE[1] and two recent competitive baselines: RotatE and TorusE. Note that QuatE[3] uses almost the same number of free parameters as QuatE[1]. TorusE uses a very large embedding dimension 10000 for both WN18 and FB15K. This number is even

Table 6: Number of free parameters comparison.

| Model | TorusE | RotatE | QuatE[1] |
|---|---|---|---|
| **Space** | $\mathbb{T}^k$ | $\mathbb{C}^k$ | $\mathbb{H}^k$ |
| **WN18** | 409.61M | 40.95M | 49.15M ($\uparrow$ 20.0%) |
| **FB15K** | 162.96M | 31.25M | 26.08M($\downarrow$ 16.5%) |
| **WN18RR** | - | 40.95M | 16.38M($\downarrow$ 60.0%) |
| **FB15K-237** | - | 29.32M | 5.82M($\downarrow$ 80.1%) |

close to the entities amount of FB15K which we think is not preferable since our original intention is to embed entities and relations to a lower dimensional space. QuatE reduces the parameter size of the complex-valued counterpart RotatE largely. This is more significant on datasets without trivial inverse relations, saving up to 80% parameters while maintaining superior performance.

**Ablation Study on Quaternion Normalization**. We remove the normalization step in QuatE and use the original relation quaternion $W_r$ to project head entity. From Table 7, we clearly observe that normalizing the relation to unit quaternion is a critical step for the embedding performance. This is likely because scaling effects in nonunit quaternions are detrimental.

**Hamilton Products between Head and Tail Entities.** We reformulate the scoring function of QuatE following the original formulate of ComplEx. We do Hamilton product between head and tail quaternions and consider the relation quaternion as weight. Thus, we have $\phi(h, r, t) = W_r \cdot (Q_h \otimes Q_t)$. As a result, the geometric property of relational rotation is lost, which leads to poor performance as shown in Table 7.

**Additional Rotational Quaternion for Tail Entity.** We hypothesize that adding an additional relation quaternion to tail entity might bring the model more representation capability. So we revise the scoring function to $(Q_h \otimes W_r^{\triangleleft}) \cdot (Q_t \otimes V_r^{\triangleleft})$, where $V_r$ represents the rotational quaternion for tail entity. From Table 7, we observe that it achieves competitive results without extensive tuning. However, it might cause some losses of efficiency.

# 6    Conclusion

In this paper, we design a new knowledge graph embedding model which operates on the quaternion space with well-defined mathematical and physical meaning. Our model is advantageous with its capability in modelling several key relation patterns, expressiveness with higher degrees of freedom as well as its good generalization. Empirical experimental evaluations on four well-established datasets show that QuatE achieves an overall state-of-the-art performance, outperforming multiple recent strong baselines, with even fewer free parameters.

**Acknowledgments**

This research was partially supported by grant ONRG NICOP N62909-19-1-2009

## Footnotes

[2]A plane of rotation is an abstract object used to describe or visualize rotations in space.

[3]https://wordnet.princeton.edu/

[4]https://pytorch.org/

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
