[Supplementary Material · NIPS_2019__QuatE_v1-12-14.pdf]

# 7 Appendix

## 7.1 Proof of Antisymmetry and Inversion

**Proof of antisymmetry pattern**    In order to prove the antisymmetry pattern, we need to prove the following inequality when imaginary components are nonzero:

$$Q_h \otimes W_r^\lhd \cdot Q_t \neq Q_t \otimes W_r^\lhd \cdot Q_h \tag{11}$$

Firstly, we expand the left term:

$$
\begin{aligned}
Q_h \otimes W_r^\lhd \cdot Q_t = & [(a_h \circ p - b_h \circ q - c_h \circ u - d_h \circ v) + (a_h \circ q + b_h \circ p + c_h \circ v - d_h \circ u)\mathbf{i} + \\
& (a_h \circ u - b_h \circ v + c_h \circ p + d_h \circ q)\mathbf{j} + (a_h \circ v + b_h \circ u - c_h \circ q + d_h \circ p)\mathbf{k}] \cdot (a_t + b_t\mathbf{i} + c_t\mathbf{j} + d_t\mathbf{k}) \\
= & (a_h \circ p - b_h \circ q - c_h \circ u - d_h \circ v) \cdot a_t + (a_h \circ q + b_h \circ p + c_h \circ v - d_h \circ u) \cdot b_t + \\
& (a_h \circ u - b_h \circ v + c_h \circ p + d_h \circ q) \cdot c_t + (a_h \circ v + b_h \circ u - c_h \circ q + d_h \circ p) \cdot d_t \\
= & \langle a_h, p, a_t \rangle - \langle b_h, q, a_t \rangle - \langle c_h, u, a_t \rangle - \langle d_h, v, a_t \rangle + \langle a_h, q, b_t \rangle + \\
& \langle b_h, p, b_t \rangle + \langle c_h, v, b_t \rangle - \langle d_h, u, b_t \rangle + \langle a_h, u, c_t \rangle - \langle b_h, v, c_t \rangle + \\
& \langle c_h, p, c_t \rangle + \langle d_h, q, c_t \rangle + \langle a_h, v, d_t \rangle + \langle b_h, u, d_t \rangle - \langle c_h, q, d_t \rangle + \langle d_h, p, d_t \rangle
\end{aligned}
$$

We then expand the right term:

$$
\begin{aligned}
Q_t \otimes W_r^\lhd \cdot Q_h = & [(a_t \circ p - b_t \circ q - c_t \circ u - d_t \circ v) + (a_t \circ q + b_t \circ p + c_t \circ v - d_t \circ u)\mathbf{i} + \\
& (a_t \circ u - b_t \circ v + c_t \circ p + d_t \circ q)\mathbf{j} + (a_t \circ v + b_t \circ u - c_t \circ q + d_t \circ p)\mathbf{k}] \cdot (a_h + b_h\mathbf{i} + c_h\mathbf{j} + d_h\mathbf{k}) \\
= & (a_t \circ p - b_t \circ q - c_t \circ u - d_t \circ v) \cdot a_h + (a_t \circ q + b_t \circ p + c_t \circ v - d_t \circ u) \cdot b_h + \\
& (a_t \circ u - b_t \circ v + c_t \circ p + d_t \circ q) \cdot c_h + (a_t \circ v + b_t \circ u - c_t \circ q + d_t \circ p) \cdot d_h \\
= & \langle a_t, p, a_h \rangle - \langle b_t, q, a_h \rangle - \langle c_t, u, a_h \rangle - \langle d_t, v, a_h \rangle + \langle a_t, q, b_h \rangle + \\
& \langle b_t, p, b_h \rangle + \langle c_t, v, b_h \rangle - \langle d_t, u, b_h \rangle + \langle a_t, u, c_h \rangle - \langle b_t, v, c_h \rangle + \\
& \langle c_t, p, c_h \rangle + \langle d_t, q, c_h \rangle + \langle a_t, v, d_h \rangle + \langle b_t, u, d_h \rangle - \langle c_t, q, d_h \rangle + \langle d_t, p, d_h \rangle
\end{aligned}
$$

We can easily see that those two terms are not equal as the signs for some terms are not the same.

**Proof of inversion pattern**    To prove the inversion pattern, we need to prove that:

$$Q_h \otimes W_r^\lhd \cdot Q_t = Q_t \otimes \bar{W}_r^\lhd \cdot Q_h \tag{12}$$

We expand the right term:

$$
\begin{aligned}
Q_t \otimes \bar{W}_r^\lhd \cdot Q_h = & [(a_t \circ p + b_t \circ q + c_t \circ u + d_t \circ v) + (-a_t \circ q + b_t \circ p - c_t \circ v + d_t \circ u)\mathbf{i} + \\
& (-a_t \circ u + b_t \circ v + c_t \circ p - d_t \circ q)\mathbf{j} + (-a_t \circ v - b_t \circ u + c_t \circ q + d_t \circ p)\mathbf{k}] \cdot (a_h + b_h\mathbf{i} + c_h\mathbf{j} + d_h\mathbf{k}) \\
= & (a_t \circ p - b_t \circ q - c_t \circ u - d_t \circ v) \cdot a_h + (a_t \circ q + b_t \circ p + c_t \circ v - d_t \circ u) \cdot b_h + \\
& (a_t \circ u - b_t \circ v + c_t \circ p + d_t \circ q) \cdot c_h + (a_t \circ v + b_t \circ u - c_t \circ q + d_t \circ p) \cdot d_h \\
= & \langle a_t, p, a_h \rangle + \langle b_t, q, a_h \rangle + \langle c_t, u, a_h \rangle + \langle d_t, v, a_h \rangle - \langle a_t, q, b_h \rangle + \\
& \langle b_t, p, b_h \rangle - \langle c_t, v, b_h \rangle + \langle d_t, u, b_h \rangle - \langle a_t, u, c_h \rangle + \langle b_t, v, c_h \rangle + \\
& \langle c_t, p, c_h \rangle - \langle d_t, q, c_h \rangle - \langle a_t, v, d_h \rangle - \langle b_t, u, d_h \rangle + \langle c_t, q, d_h \rangle + \langle d_t, p, d_h \rangle
\end{aligned}
$$

We can easily check the equality of these two terms.

## 7.2 Hyperparameters Settings

We list the best hyperparameters setting of QuatE on the benchmark datasets:

Hyperparameters for **QuatE**[1] without type constraints:

- WN18: $k = 300, \lambda_1 = 0.05, \lambda_2 = 0.05, \#neg = 10$
- FB15K: $k = 200, \lambda_1 = 0.05, \lambda_2 = 0.05, \#neg = 10$
- WN18RR: $k = 100, \lambda_1 = 0.1, \lambda_2 = 0.1, \#neg = 1$
- FB15K-237: $k = 100, \lambda_1 = 0.3, \lambda_2 = 0.3, \#neg = 10$

Figure 2: Fano Plane, a mnemonic for the products of the unit octonions.

Hyperparameters for **QuatE**[2] with N3 regularization and reciprocal learning, without type constraints:

- WN18: $k = 1000, reg = 0.05$

- FB15K: $k = 1000, reg = 0.0025$

- WN18RR: $k = 1000, reg = 0.1$

- FB15K-237: $k = 1000, reg = 0.05$

Hyperparameters for **QuatE**[3] with type constraint:

- WN18: $k = 250, \lambda_1 = 0.05, \lambda_2 = 0, \#neg = 10$

- FB15K: $k = 200, \lambda_1 = 0.1, \lambda_2 = 0, \#neg = 20$

- WN18RR: $k = 100, \lambda_1 = 0.1, \lambda_2 = 0.1, \#neg = 1$

- FB15K-237: $k = 100, \lambda_1 = 0.2, \lambda_2 = 0.2, \#neg = 10$

**Number of epochs.** The number of epochs needed of QuatE and RotatE are shown in Table 8.

Table 8: Number of epochs needed of **QuatE**[1] and RotatE.

| Datasets | WN18 | WN18RR | FB15K | FB15K-237 |
|---|---|---|---|---|
| **QuatE**[1] | 3000 | 40000 | 5000 | 5000 |
| RotatE | 80000 | 80000 | 150000 | 150000 |

### 7.3 Octonion for Knowledge Graph embedding

Apart from Quaternion, we can also extend our framework to Octonions (hypercomplex number with one real part and seven imaginary parts) and even Sedenions (hypercomplex number with one real part and fifteen imaginary parts). Here, we use **OctonionE** to denote the method with Octonion embeddings and details are given in the following text.

Octonions are hypercomplex numbers with seven imaginary components. The Octonion algebra, or Cayley algebra, $\mathbb{O}$ defines operations between Octonion numbers. An Octonion is represented in the form: $O_1 = x_0 + x_1\mathbf{e}_1 + x_2\mathbf{e}_2 + x_3\mathbf{e}_3 + x_4\mathbf{e}_4 + x_5\mathbf{e}_5 + x_6\mathbf{e}_6 + x_7\mathbf{e}_7$, where $\mathbf{e}_1, \mathbf{e}_2, \mathbf{e}_3, \mathbf{e}_4, \mathbf{e}_5, \mathbf{e}_6, \mathbf{e}_7$ are imaginary units which re the square roots of $-1$. The multiplication rules are encoded in the Fano Plane (shown in Figure 2). Multiplying two neighboring elements on a line results in the third element on that same line. Moving with the arrows gives a positive answer and moving against arrows gives a negative answer.

The conjugate of Octonion is defined as: $\bar{O}_1 = x_0 - x_1\mathbf{e}_1 - x_2\mathbf{e}_2 - x_3\mathbf{e}_3 - x_4\mathbf{e}_4 - x_5\mathbf{e}_5 - x_6\mathbf{e}_6 - x_7\mathbf{e}_7$.

The norm of Octonion is defined as: $|O_1| = \sqrt{x_0^2 + x_1^2 + x_2^2 + x_3^2 + x_4^2 + x_5^2 + x_6^2 + x_7^2}$.

If we have another Octonion: $O_2 = y_0 + y_1\mathbf{e}_1 + y_2\mathbf{e}_2 + y_3\mathbf{e}_3 + y_4\mathbf{e}_4 + y_5\mathbf{e}_5 + y_6\mathbf{e}_6 + y_7\mathbf{e}_7$. We derive the multiplication rule with the Fano Plane.

$$
\begin{aligned}
O_1 \otimes O_2 = &(x_0y_0 - x_1y_1 - x_2y_2 - x_3y_3 - x_4y_4 - x_5y_5 - x_6y_6 - x_7y_7) \\
&+ (x_0y_1 + x_1y_0 + x_2y_3 - x_3y_2 + x_4y_5 - x_5y_4 - x_6y_7 + x_7y_6)\mathbf{e}_1 \\
&+ (x_0y_2 - x_1y_3 + x_2y_0 + x_3y_1 + x_4y_6 + x_5y_7 - x_6y_4 - x_7y_5)\mathbf{e}_2 \\
&+ (x_0y_3 + x_1y_2 - x_2y_1 + x_3y_0 + x_4y_7 - x_5y_6 + x_6y_5 - x_7y_4)\mathbf{e}_3 \\
&+ (x_0y_4 - x_1y_5 - x_2y_6 - x_3y_7 + x_4y_0 + x_5y_1 + x_6y_2 + x_7y_3)\mathbf{e}_4 \\
&+ (x_0y_5 + x_1y_4 - x_2y_7 + x_3y_6 - x_4y_1 + x_5y_0 - x_6y_3 + x_7y_2)\mathbf{e}_6 \\
&+ (x_0y_6 + x_1y_7 + x_2y_4 - x_3y_5 - x_4y_2 + x_5y_3 + x_6y_0 - x_7y_1)\mathbf{e}_5 \\
&+ (x_0y_7 - x_1y_6 + x_2y_5 + x_3y_4 - x_4y_3 - x_5y_2 + x_6y_1 + x_7y_0)\mathbf{e}_7
\end{aligned}
\tag{13}
$$

We can also consider Octonions as a combination of two Quaternions. The scoring functions of OctonionE remains the same as QuatE.

$$
\phi(h, r, t) = Q_h \otimes W_r^{\triangleleft} \cdot Q_t : \{Q_h, W_r, Q_t \in \mathbb{O}^k\}
\tag{14}
$$

The results of OctonionE on dataset WN18 and WN18RR are given below. We observe that OctonionE performs equally to QuatE. It seems that extending the model to Octonion space does not give additional benefits. Octonions lose some algebraic properties such as associativity, which might bring some side effects to the model.

Table 9: Results of Octonion Knowledge graph embedding.

| | WN18 | | | | |
|---|---|---|---|---|---|
| Model | MR | MRR | Hit@10 | Hit@3 | Hit@1 |
| **OctonionE** | 182 | 0.950 | 0.959 | 0.954 | 0.944 |
| | WN18RR | | | | |
| Model | MR | MRR | Hit@10 | Hit@3 | Hit@1 |
| **OctonionE** | 2098 | 0.486 | 0.582 | 0.508 | 0.435 |