[Reviews · NeurIPS 2019]

Reviewer 1



This paper proposes a knowledge graph embedding model where entities and relations are represented in the quaternion space - this generalises some recent trends in knowledge graph embeddings where the complex space was being used to overcome the limitations of some existing models [1, 2]. The dependencies between latent features are modeled via the Hamilton product, which seems to encourage more compact interactions between latent features. The model design is sound and, with little added complexity in comparison with other methods (a slightly more convoluted scoring function, and an uncommon initialisation scheme) the model achieves robust results in comparison with other methods in the literature (see Tab. 4 - although a bit far from SOTA [2]) and the resulting models are fairly parameter efficient (see Tab. 6). Small issue: "euclidean" should be "Euclidean". [1] proceedings.mlr.press/v48/trouillon16.pdf [2] https://arxiv.org/abs/1806.07297

Reviewer 2



This paper introduces quaternion embeddings for entities and relations in a knowledge graph. Quaternions are vectors in hypercomplex space with 3 imaginary components (2 more than usual complex numbers). It also proposes to use the Hamilton product to score the existence of a triple. Hamilton product are non-commutative which is highly desirable property. Also Hamilton product, represents scaling and smooth rotation and are easier to compose and avoid the problem of gimbal lock. They are also numerically stable when compared to rotational matrices. The score function is s(e1, r, e2) is defined first by a Hamilton product between e1 and normalized r (to only model rotation), followed by a inner product with e2 to output a scalar score. The paper also shows that both complex embeddings and distmult are special cases of quaternion embeddings. The paper also gets convincing results on benchmark datasets such as FB15k-237 and WN18RR. The ablation tests are convincing and it has comparable number of parameters wrt the baseline. Originality: The paper presents a novel way of representing entities and relations Clarity: The paper is well-written and was easy to follow. Significance: the results are significance, and the paper will be impactful.

Reviewer 3



After rebuttal: Thanks for providing more results. That addressed my 3rd concern to some extent (although it seems like your model requires many more epochs compared to ComplEx and I wonder how ComplEx will perform given the same number of epochs and using uniform negative sampling, but this is not a major concern). I'm not yet convinced about the issue I raised regarding relation normalization though. "We also found that the relation normalization can improve the ComplEx model as well. But it is till worse than QuatE." Why not provide some actual numbers similar to the other cases so we can see how much better QuatE is compared to ComplEx when they both use relation normalization? And in case QuatE actually outperforms ComplEx significantly in presence of relation normalization, what is special about QuatE compared to ComplEx that makes it benefit from relation normalization to the extent that QuatE + relation normalization outperforms ComplEx + relation normalization, while ComplEx without relation normalization outperforms QuatE without relation normalization? =============================== Knowledge graph completion has been vastly studied during the past few years with more than 30 embedding-based approaches proposed for the task. Inspired by the recent success of ComplEx and RotatE in using embeddings with complex numbers, this paper proposes to use quaternion embeddings instead of complex embeddings. 1- Novelty: My major reservation for this work concerns the novelty. While going from real-valued embeddings to complex-valued embeddings was novel and addressed some of the issues with simple approaches based on real-valued embeddings (e.g., they addressed the symmetry issue of DistMult), going beyond complex-valued embeddings may not be super novel. This is especially concerning as I think with the same number of parameters, ComplEx may work quite competitive with QuatE. 2- My second concern is regarding one of the experiments in the ablation study. According to Table 7, when the authors remove the relation normalization, then the performance of QuatE drops significantly showing (in three cases) sub-par results even compared to ComplEx. This raises the question that maybe the performance boost is only coming from relation normalization and not from using quaternions. 3- The results may be a bit unfair towards previous approaches. The authors report the results of several of the previous models directly from their papers while they use a somewhat different experimental setup. In particular, the embedding dimensions and the number of negative examples the authors generate per positive example are (for some datasets) larger than those used in previous papers. Also, the authors use a different way of generating negative examples (uniform sampling) rather than the corruption method introduced by Bordes et al. 2013 and used by the reported baselines. 4- How many epochs does it take your model to train? I'm assuming with a uniform negative sampling, it should take many epochs. Is this correct? 5- I may be missing something but the results in Table 4 and 5 do not seem to match. According to table 5, for WN18PR, QuatE outperforms RotatE on most relations with a large margin. However, the overall MRR of QuatE on this dataset is only slightly better than RotatE. Why is this happening? Minor comments: 6- The correct reference for the proof of isomorphism between ComplEx and HolE is https://arxiv.org/pdf/1702.05563.pdf 7- "Compared to Euler angles, quaternion can avoid the problem of gimbal lock" requires much more explanation. 8- In line 172, I think "antisymmetric" should be replaced with "asymmetric" 9- I liked the criticism on fixing the composition patterns. 10- The paragraph on "connection to DistMult and ComplEx" is kind of obvious and not necessary,

Reviewer 4



Using quaternions seems novel. The experimental results look good too. However, the necessity of quaternions is not very clear, i.e., why more degrees of freedom is needed? And, do quaternions solve the problem in the example of "elder brother"? In the appendix, octonions are considered, but similarly, there is almost no explanation why this is needed. In the conclusion, "good generalization" is mentioned as an advantage, but I did not see any corresponding rigorous argument in this paper.

[Author Response · NeurIPS 2019]

Thanks to all reviewers for your constructive suggestions. Responses are as follows.

Major characteristics/advantages of the proposed approach:

• QuatE considers relations as rotations in four dimensional space. It firstly rotates the head entities then do
semantic matching between the rotated head entity and the tail entity. QuatE is a generalization of ComplEx,
it keeps all the benefits of ComplEx. We showed that quaternion rotations are especially helpful for the
knowledge graph embedding.

• It can greatly save the number of parameters. This is more significant on datasets without trivial inverse
relations. For example, it reduced the number of parameters by 80.1% on FB15K-237, 60% on WN18RR,
compared to the latest state-of-the-art model(RotatE).

**Comparison with ComplEx by controlling the number of**
**parameters and negative samples.** For datasets WN18RR
and FB15K-237, the reported results of ComplEx are achieved
with embeddings size 200 while QuatE use embedding size
100. The numbers of parameters are the same, but QuatE
outperforms ComplEx largely. We also ran ComplEx on WN18
using the same number of parameters and negative samples as
QuatE. As shown in Table 1, QuatE still performs better than
ComplEx.

Table 1: Results of ComplEx and QuatE with same number of parameters and negative samples.

| | WN18 | | WN18RR | |
|---|---|---|---|---|
| #Params | 40.96M | | 16.38M | |
| #neg | 10 | | 1 | |
| Measures | MRR | Hit@10 | MRR | Hit@10 |
| ComplEx | 0.942 | 0.952 | 0.44 | 0.51 |
| QuatE | 0.950 | 0.959 | 0.488 | 0.582 |

**Most baselines are exhaustively tuned.** The hyper-
parameters of baselines are already exhaustively tuned. For
example, the number of negative samples in the original Com-
plEx model are tuned from {1, 2, 5, 10}. Some neural network-based methods even use dropout and label smoothing to
improve their performance. For QuatE, the number of negative samples are 10(WN18), 20 (FB15K), 1(WN18RR),
10(FB15K-237). The size is fair compared with ComplEx. If we set #neg=10 for FB15K, we can get MRR=0.781,
Hit@10=0.899.

**Number of epochs.** The number of epochs needed of
QuatE and RotatE are shown in Table 2, despite that
we use uniform sampling, and rotatE use adversarial
negative sampling, our method needs much less number
of epochs than RotatE.

Table 2: Number of epochs needed of QuatE and RotatE.

| Datasets | WN18 | WN18RR | FB15K | FB15K-237 |
|---|---|---|---|---|
| QuatE | 1500 | 40000 | 5000 | 15000 |
| RotatE | 80000 | 80000 | 150000 | 150000 |

**Discussion on the composition patterns**. Composi-
tion patterns are commonplaces in knowledge graphs.
Here, we pointed out that fixing the composition function may lead to sub-optimal performances as there are many ways
of relation compositions. Our model does not fix the composition pattern of the model. If $r_3$ composes of $r_1$ and $r_2$,
both TransE and RotatE assume there are only one determinate composition functions ($r_3 = r_1 + r_2$ or $r_3 = r_1 \circ r_2$).
In these two models, $r_3$ has nothing to do with the entities. In QuatE, the $r_3$ is not only determined by relations $r_1$ and
$r_2$, but also the entity embeddings. As such, the composition patterns are not fixed to one form, instead, relation $r_3$ is
not only determined by $r_1$ and $r_2$ but also simultaneously influenced by entity embeddings.

**MRR for each relation on WN18RR.** The overall MRR improvement on WN18RR is 0.470 ->0.488. QuatE get
improvements on seven relations, and are on par or fail on other relations. Note that the number of samples for each
relation is different. Thus the overall improvement is weighted by the number of samples of each relation.

We also found that the relation normalization can improve the ComplEx model as well. But it is till worse than QuatE.
In this ablation study, we did not tune the hyper-parameters but using the same ones as standard QuatE. After some
tests, we found that the initialization scheme is optional on these four datasets, random initialization can get the same
performance. This initialization scheme might be useful for other datasets.

[Meta-Review · NeurIPS 2019]

The paper attempts learn better entity and relation embeddings for knowledge graphs. In this regard, the authors employ quarternion algebra with Hamilton product, which is used as the scoring function for knowledge triplets. Hamilton product is asymmetric, which is claimed to be beneficial for modeling directed egdes in a knowledge graph. Further the paper outperforms many well established methods and the authors seem to have done an exhaustive set of experiments. However, all the reviewers are in consensus that motivation for the use of quarternions is not clear, e.g. the paper does a poor job in demonstrating how does more degrees of freedom in rotation help in learning better embedding. This should be made clear in the camera ready version. During discussion, one possible direction of explanation came up which you might pursue further: QuatE learns a representation of relations which are maximally orthogonal to head/tail entities representations. This is in contrast to most other methods that learn relations which are like a difference between head and tail representations. Such hypothesis can be easily verified empirically as well. Looking at connections of triple products and quarternions products might be a starting place. Adding such insights would improve the paper significance a lot. So please put in a genuine effort in working out a motivation for the camera ready version. Also in empirical comparison with complex, please include exact numbers with and without relation normalization in the final version. Thus, I am barely recommending acceptance to NeurIPS.